# Event-Driven Dynamic Scene Depth Completion

**Zhiqiang Yan**[1]    **Jianhao Jiao**[2]    **Zhengxue Wang**[3]    **Gim Hee Lee**[1]

[1]National University of Singapore    [2]University College London
[3]Nanjing University of Science and Technology
{yanzq, gimhee.lee}@nus.edu.sg

## Abstract

Depth completion in dynamic scenes poses significant challenges due to rapid ego-motion and object motion, which can severely degrade the quality of input modalities such as RGB images and LiDAR measurements. Conventional RGB-D sensors often struggle to align precisely and capture reliable depth under such conditions. In contrast, event cameras with their high temporal resolution and sensitivity to motion at the pixel level provide complementary cues that are beneficial in dynamic environments. To this end, we propose **EventDC**, the first event-driven depth completion framework. It consists of two key components: Event-Modulated Alignment (EMA) and Local Depth Filtering (LDF). Both modules adaptively learn the two fundamental components of convolution operations: offsets and weights conditioned on motion-sensitive event streams. In the encoder, EMA leverages events to modulate the sampling positions of RGB-D features to achieve pixel redistribution for improved alignment and fusion. In the decoder, LDF refines depth estimations around moving objects by learning motion-aware masks from events. Additionally, EventDC incorporates two loss terms to further benefit global alignment and enhance local depth recovery. Moreover, we establish the first benchmark for event-based depth completion comprising one real-world and two synthetic datasets to facilitate future research. Extensive experiments on this benchmark demonstrate the superiority of our EventDC. Project page.

## 1 Introduction

Depth completion [48, 33, 36, 47, 65] aims to predict dense depth from sparse measurements, typically using auxiliary modalities such as RGB images. As a cornerstone of 3D perception, it plays a crucial role in a wide range of downstream applications including self-driving [70, 57, 26, 52], augmented reality [45, 60, 54, 66, 71], scene understanding [56, 40, 78, 53, 72], *etc*. Although recent methods have demonstrated impressive results in static scenes, dynamic environments remain highly challenging. As illustrated in Fig. 1(a), the rapid ego-motion results in blurry RGB images and misalignment with LiDAR measurements, while fast-moving objects further exacerbate depth inaccuracies in their vicinity. These challenges make precise depth completion even more difficult.

The unique characteristics of event cameras [10, 9, 11] provide a compelling complement to conventional RGB-D sensors in dynamic scenes. Their microsecond-level temporal resolution enables the reliable capture of rapid ego-motion without introducing motion blur, and their asynchronous change-driven operation makes them inherently well-suited for detecting fast-moving objects. These properties help mitigate the limitations of RGB-D measurements by offering temporally consistent and low-latency signals, particularly in regions where traditional sensors often fail. As a result, event-based sensing proves especially advantageous for depth completion in highly dynamic environments.

In this work, we present EventDC, a novel depth completion framework that leverages event data to tackle the challenges posed by dynamic scenes. As shown in Fig. 1(b), the core idea is to exploit the unique properties of event streams to guide depth completion especially in motion-affected regions.

39th Conference on Neural Information Processing Systems (NeurIPS 2025).

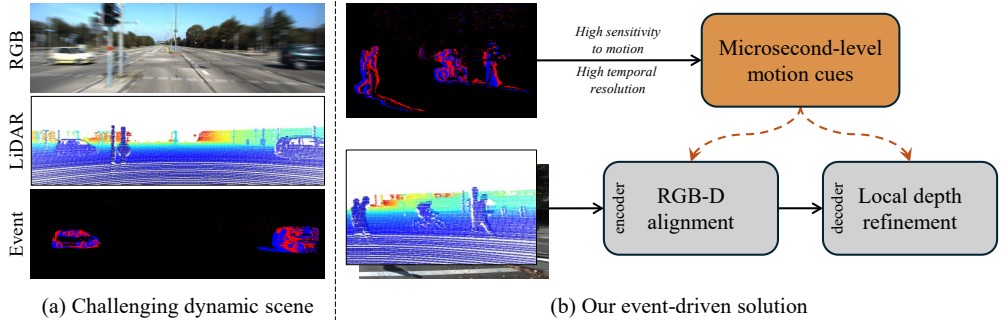

(a) Challenging dynamic scene          (b) Our event-driven solution

Figure 1: Data example and our solution for depth completion in dynamic environments. Leveraging high temporal resolution and motion sensitivity, event provides valuable complementary information for depth completion in dynamic scenes. Multiple event streams are aggregated for clear visualization.

To this end, our EventDC incorporates two key components: Event-Modulated Alignment (EMA) and Local Depth Filtering (LDF). EMA is an encoder-side module that adaptively adjusts convolutional sampling positions using event information to achieve pixel redistribution for enhanced global alignment and more effective multi-modal fusion between RGB and LiDAR features. Furthermore, it incorporates a structure-aware loss to mitigate the RGB-D inconsistency caused by rapid ego-motion. LDF is a decoder-side module that focuses on refining depth around moving objects. It first learns motion masks from event streams to get the regions influenced by object motion. The learned masks are then used by LDF with a local motion-aware constraint to facilitate more accurate depth predictions in these regions. Concurrently, the two modules enable our EventDC to address both global misalignment and local depth inaccuracies for handling complex scenarios involving motion.

Additionally, depth completion based on event cameras remains an underexplored area with no existing event-based depth completion datasets to date. To address this gap, we introduce the first event-based depth completion benchmark, which includes a real-world dataset *EventDC-Real*, a semi-synthetic dataset *EventDC-SemiSyn*, and a fully synthetic dataset *EventDC-FullSyn*.

In summary, our contributions are as follows:

- To the best of our knowledge, we are the first to introduce EventDC, a novel event-driven depth completion framework designed to address the challenges of dynamic environments.

- We present two event-driven modules: EMA and LDF which are designed to mitigate the global misalignment caused by ego-motion and local depth inaccuracies due to object motion. Additionally, these two modules are jointly supported by two dedicated loss constraints.

- To foster further research, we build the first event-based benchmark for depth completion. Extensive experiments across these datasets demonstrate the superiority of our approach, with up to 12.8% improvement on the best-performing dataset over suboptimal methods.

## 2 Related Work

**Depth Completion.** Early depth completion methods [48, 33, 25, 7, 49, 32] focus on predicting dense depth maps directly from sparse inputs. For example, IP-Basic [25] uses traditional image processing techniques to densify sparse depth without deep learning. In contrast, Uhrig et al. [48] introduce Sparsity Invariant CNNs, which adapt convolutional operations to varying input densities to ensure consistent performance. S2D [33] employs an encoder-decoder architecture to progressively densify sparse depth input. FusionNet [49] integrates global context and local structures with a confidence-driven refinement mechanism. Eldesokey et al. [7] present a confidence propagation method within CNNs to improve sparse depth regression by modeling uncertainty. Guided depth completion using color images has gained significant traction [46, 70, 65, 20, 69, 73, 47, 63, 64]. Dynamic filtering techniques [46, 61, 62] generate adaptive filtering kernels from color images for effective extraction of depth features. Methods such as FuseNet [1], PointDC [67], BEVDC [73], and TPVD [65] further enhance depth completion by incorporating raw point clouds. Moreover, priors of the depth foundation models are used to improve generalization [38, 37, 51, 16]. Recently, SigNet [66] redefines depth completion as enhancement, densifying sparse depth with non-CNN methods, and then refines

it through a degradation-aware framework. In addition, SPN techniques [3, 4, 36, 59, 20, 28, 65], which serve as effective refinement modules, can further enhance performance.

**Event-Based Depth Estimation.** Depth estimation with event cameras [9, 15, 10, 8, 13, 17, 30, 58] attracts growing interest due to the high temporal resolution, dynamic range, and low latency of asynchronous vision sensors. Early methods reconstruct depth solely from event streams such as the end-to-end framework by Hidalgo-Carrió et al. [17], the multi-view stereo pipeline EMVS [39], and unsupervised learning approaches for depth and egomotion [74]. DERD-Net [18] further exploits 3D convolutions and recurrence on event-based disparity space images, and Zhu et al. [77] propose a self-supervised framework for joint depth and optical flow estimation. Recent works leverage additional modalities to enhance event-based depth estimation. EMoDepth [75] temporally aligns events and intensity frames for self-supervised monocular depth learning. Muglikar et al. [34] propose event-guided illumination control for active depth sensors. SRFNet [35] fuses frame and event features for fine-grained depth prediction with improved structure in both daytime and nighttime scenes. SDT [68] combines spiking neural networks and transformers for efficient depth estimation. Furthermore, contrast maximization that emerges as a fundamental principle for event-based motion, depth, and optical flow estimation [9, 41] has inspired many subsequent works.

**Dynamic Convolution.** Dynamic convolution is a method that adjusts the convolution operation based on input features, and it gains significant attention in computer vision tasks. Techniques such as graph convolution and deformable convolution serve as specific manifestations of dynamic convolution. For example, ACMNet and GraphCSPN build graph structures to enable effective multi-modal fusion and refinement. STN [21] introduces the concept of spatially transforming features within a network, although training such a mechanism is a challenging task. Following this, DFN [23] proposes an approach that adapts filter parameters based on input features despite maintaining fixed kernel sizes. Deformable Convolution [5, 76] takes a different approach with the focus on dynamically adjusting sampling locations by generating offsets based on the geometric properties of objects. Similarly, Active Convolution [22] improves sampling by adjusting the locations while keeping the kernel shape fixed. More recently, GuideNet [46] develops a guided convolution block specifically designed for multi-modal data. Despite these innovations, dynamic mechanisms often add considerable complexity. To address this issue, RigNet [61] simplifies the dynamic guidance process by employing convolution factorization combined with attention [19].

## 3 Our Method

### 3.1 Background

The core of dynamic convolution lies in the adaptive determination of sampling positions and weights. Graph Convolutional Networks (GCNs) [24, 50] and Deformable Convolutional Networks (DCNs) [5, 76] serve as representative implementations of this concept. GCNs define sampling locations as neighboring nodes within the graph structure and compute adaptive weights during the aggregation stage. On the other hand, DCNs determine sampling locations through learned offsets and obtain adaptive weights by modulating predefined kernel weights with learned scalars. Both GCNs and DCNs can be viewed as extensions of standard convolutional operations, where the sampling locations and weights are made learnable and structure-aware. We use DCNs as an example to illustrate this dynamic learning process.

Specifically, DCNv1 [5] introduces learnable offsets for each sampling location to shift adaptively. Subsequently, DCNv2 [76] further incorporates a learnable modulation scalar for each sampling position that enables the assignment of varying importance to different locations. Given an input feature map $\mathbf{x}$ and a convolutional kernel with $K$ sampling positions, let $\mathbf{w}_k$ and $\mathbf{p}_k$ denote the weight and the pre-defined offset of the $k$-th position, respectively. DCNv2 can be formulated as:

$$\hat{\mathbf{x}}(\mathbf{p}_0) = \sum_{k=1}^{K} \mathbf{w}_k \cdot \mathbf{x}(\mathbf{p}_0 + \mathbf{p}_k + \Delta\mathbf{p}_k) \cdot \Delta\mathbf{m}_k, \tag{1}$$

where $\mathbf{p}_0$ denotes the reference location, and $\Delta\mathbf{p}_k$ and $\Delta\mathbf{m}_k$ are the learnable offset and modulation scalar, respectively. Note that $\mathbf{w}_k$ and $\Delta\mathbf{m}_k$ can be jointly interpreted as a unified learnable term. As a result, the adaptive adjustment of offset and weight in DCNv2 provides a foundation for leveraging event data to tackle the challenges posed by fast motion in depth completion.

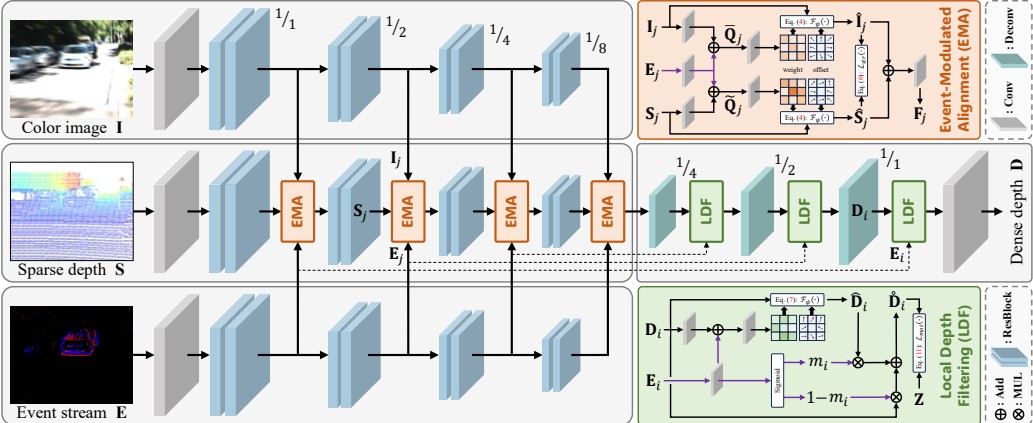

Figure 2: Pipeline of our EventDC. The color image $\mathbf{I}$, sparse depth $\mathbf{S}$, and event stream $\mathbf{E}$ are first processed by three structurally identical encoders. At each stage, the Event-Modulated Alignment (EMA) block leverages event features to align and fuse RGB-D representations. In the decoder, the Local Depth Filtering (LDF) unit further enhances depth estimation around moving objects, guided by the inherent sensitivity of events to motion and reinforced by local motion-aware constraints.

## 3.2 EventDC Architecture

**Overview.** In highly dynamic environments, the proposed approach is designed to alleviate the adverse effects of fast motion that include global misalignment and local depth inaccuracies caused by ego-motion and object motion, respectively. Fig. 2 illustrates the pipeline of our EventDC, which begins by employing three structurally consistent encoders to extract features from the color image $\mathbf{I}$, sparse depth $\mathbf{S}$, and event data $\mathbf{E}$. This yields multi-scale representations $\{\mathbf{I}_1, \mathbf{I}_2, \mathbf{I}_3, \mathbf{I}_4\}$, $\{\mathbf{S}_1, \mathbf{S}_2, \mathbf{S}_3, \mathbf{S}_4\}$, and $\{\mathbf{E}_1, \mathbf{E}_2, \mathbf{E}_3, \mathbf{E}_4\}$ at the $\{^1/_1, ^1/_2, ^1/_4, ^1/_8\}$ stages, respectively. In the decoder, three deconvolution layers are applied to progressively generate $\{\mathbf{D}_3, \mathbf{D}_2, \mathbf{D}_1\}$ at the $\{^1/_4, ^1/_2, ^1/_1\}$ stages, respectively. Furthermore, EventDC incorporates two key components: Event-Modulated Alignment (EMA) and Local Depth Filtering (LDF). At each encoder stage, EMA predicts spatial offsets from event features and uses them to adjust the pixel distributions of RGB and depth features. This enables more precise multi-modal alignment and fusion. In addition, a structure-aware loss is introduced to further enhance the consistency. At the decoder stage, LDF leverages event features to estimate motion masks that identify moving objects. It then refines the depth values within these regions using dynamic convolutions and a local motion-aware loss, ultimately enhancing depth accuracy around the moving objects.

**Event-Modulated Alignment.** As depicted in Fig. 2, at the $j$-th ($j \in \{1, 2, 3, 4\}$) stage of the three encoders, the EMA module takes as input the color image feature $\mathbf{I}_j$, sparse depth feature $\mathbf{S}_j$, and event feature $\mathbf{E}_j$, each with dimensions $\mathbb{R}^{C \times H \times W}$, where $C$, $H$, and $W$ denote the channel, height, and width, respectively. These inputs are first individually processed by three separate $3 \times 3$ convolutional layers $\mathcal{F}_{\tau_{j1}}(\cdot)$, $\mathcal{F}_{\tau_{j2}}(\cdot)$ and $\mathcal{F}_{\tau_{j3}}(\cdot)$, with a stride of 1 and output channels of $C$, $2C$, and $C$, respectively. The transformed event feature is then fused with the transformed RGB and depth features, respectively, resulting in the intermediate features:

$$\bar{\mathbf{Q}}_j = \mathcal{F}_{\tau_{j1}}(\mathbf{I}_j) + \alpha \cdot \mathcal{F}_s\left(\mathcal{F}_{\tau_{j2}}(\mathbf{E}_j)\right), \tag{2a}$$

$$\tilde{\mathbf{Q}}_j = \mathcal{F}_{\tau_{j3}}(\mathbf{S}_j) + \beta \cdot \mathcal{F}_s\left(\mathcal{F}_{\tau_{j2}}(\mathbf{E}_j)\right), \tag{2b}$$

where $\mathcal{F}_s(\cdot)$ denotes the operation that splits the $2C$-channel feature into two $C$-channel parts. $\alpha$ and $\beta$ are learnable terms [1] that control the contribution of the event term.

Subsequently, these two intermediate features are used to predict the offsets via two additional $3 \times 3$ convolutions, $\mathcal{F}_{\tau_{j4}}(\cdot)$ and $\mathcal{F}_{\tau_{j5}}(\cdot)$, producing $2K \times H \times W$ offsets and $K \times H \times W$ weights:

$$\Delta \bar{\mathbf{p}}_j, \bar{\mathbf{w}}_j = \mathcal{F}_{\tau_{j4}}(\bar{\mathbf{Q}}_j), \tag{3a}$$

$$\Delta \tilde{\mathbf{p}}_j, \tilde{\mathbf{w}}_j = \mathcal{F}_{\tau_{j5}}(\tilde{\mathbf{Q}}_j). \tag{3b}$$

---

[1] Implemented using `torch.nn.Parameter(zeros(1))` with the zero initialization designed to facilitate the progressive learning of event priors during training.

This step enables the model to adaptively determine the sampling locations by learning a prior from event data that is sensitive to fast motion. Consequently, the dynamic convolution in Eq. (1) can be used to perform pixel-wise adjustment of image and depth features formulated as follows:

$$\hat{\mathbf{I}}_j = \mathcal{F}_\psi(\mathbf{I}_j;\ \Delta\bar{\mathbf{p}}_j, \bar{\mathbf{w}}_j), \tag{4a}$$

$$\hat{\mathbf{S}}_j = \mathcal{F}_\psi(\mathbf{S}_j;\ \Delta\tilde{\mathbf{p}}_j, \tilde{\mathbf{w}}_j), \tag{4b}$$

where $\mathcal{F}_\psi(\cdot)$ denotes the generalized form of the operation defined in Eq. (1). Note that Eq.(4) emphasizes RGB-D pixel redistribution under highly dynamic conditions with offsets instead of adaptive weights. Consequently, in contrast to Eq.(1), both $\bar{\mathbf{w}}_j$ and $\tilde{\mathbf{w}}_j$ correspond to the predefined weights $\mathbf{w}$ with $\Delta\mathbf{m}$ being the identity matrix. Subsequently, the redistributed RGB-D features are further processed by a $3 \times 3$ convolution $\mathcal{F}_{\tau_{j6}}(\cdot)$ to obtain the fused feature $\mathbf{F}_j \in \mathbb{R}^{C \times H \times W}$, which is formulated as:

$$\mathbf{F}_j = \mathcal{F}_{\tau_{j6}}(\hat{\mathbf{I}}_j + \hat{\mathbf{S}}_j). \tag{5}$$

Additionally, a structure-aware loss $\mathcal{L}_{str}$ is introduced to enhance the consistency. Let $\mathcal{G}(\cdot)$ denote a sequence of single-channel convolution, Min-Max normalization, and gradient computation:

$$\mathcal{L}_{str} = \sum_{j=1}^{4} \frac{1}{n} \|\mathcal{G}(\hat{\mathbf{I}}_j) - \mathcal{G}(\hat{\mathbf{S}}_j)\|_2^2. \tag{6}$$

**Local Depth Filtering.** As shown in Fig.2, the LDF module takes the depth feature $\mathbf{D}_i$ and event feature $\mathbf{E}_i$ as inputs to adaptively generate offsets and weights at the $i$-th stage of the decoder ($i \in \{1, 2, 3\}$). Following the strategy used in Eqs.(2)–(4), this results in the updated depth feature:

$$\hat{\mathbf{D}}_i = \mathcal{F}_\psi(\mathbf{D}_i;\ \Delta\tilde{\mathbf{p}}_i, \tilde{\mathbf{w}}_i). \tag{7}$$

In contrast, the modulation scalar $\Delta m$ within $\tilde{\mathbf{w}}_i$ is learned jointly from the depth and event inputs. Furthermore, to explicitly model regions of dynamic objects, LDF predicts a motion mask $m_i$ based on $\mathbf{E}_i$ using a sigmoid activation $\sigma(\cdot)$ after a single-channel $3 \times 3$ convolution $\mathcal{F}_{\tau_{i6}}(\cdot)$:

$$m_i = \sigma(\mathcal{F}_{\tau_{i6}}(\mathbf{E}_i)). \tag{8}$$

By combining Eqs. (7) and (8), LDF refines depth with a focus on dynamic regions to get:

$$\mathring{\mathbf{D}}_i = m \cdot \hat{\mathbf{D}}_i + (1 - m) \cdot \mathbf{D}_i. \tag{9}$$

Finally, the output $\mathring{\mathbf{D}}_1$ from the last LDF module is passed through a $3 \times 3$ convolutional tail $\mathcal{F}_{\tau_t}(\cdot)$ to generate the dense depth prediction:

$$\mathbf{D} = \mathcal{F}_{\tau_t}(\mathring{\mathbf{D}}_1). \tag{10}$$

Additionally, we introduce a motion-aware loss to enhance the depth recovery around motion areas:

$$\mathcal{L}_{mot} = \sum_{i=1}^{3} \frac{1}{n} \|b_i \cdot \mathcal{H}(\mathring{\mathbf{D}}_i) - b_i \cdot \mathcal{F}_d(\mathbf{Z})\|_2^2, \tag{11}$$

where $\mathbf{Z}$ is the GT depth, $\mathcal{H}(\cdot)$ applies ReLU and a single-channel convolution, and $b_i$ is a binary mask with $b_i = 1$ if $m_i$ exceeds its mean, and 0 otherwise. $\mathcal{F}_d(\cdot)$ denotes the downsampling operation.

**Discussion.** In summary, EMA and LDF differ from previous dynamic convolution methods in two key aspects: **(1)** Unlike traditional methods that typically rely on single-modal and single-path inputs, our approach adopts a multi-modal and multi-path input design, where key convolutional parameters are derived from different modalities. **(2)** Our method is data-driven where we use event-based adaptation to address global misalignment and local depth inaccuracies caused by fast motion.

### 3.3 Loss Function

Given the predicted depth $\mathbf{D}$ and GT depth $\mathbf{Z}$ with $n$ valid pixels, we adopt a commonly used reconstruction loss [36, 27, 69, 56, 55] to formulate the training objective:

$$\mathcal{L}_{rec} = \frac{1}{n}(\|\mathbf{D} - \mathbf{Z}\|_2^2 + \|\mathbf{D} - \mathbf{Z}\|_1). \tag{12}$$

By combining the reconstruction loss with the structure-constrained loss $\mathcal{L}_{str}$ in Eq. 6 and motion-aware loss $\mathcal{L}_{rec}$ in Eq. 11, the overall loss function is formulated as:

$$\mathcal{L}_t = \mathcal{L}_{rec} + \lambda\mathcal{L}_{str} + \mu\mathcal{L}_{mot}, \tag{13}$$

where $\lambda$ and $\mu$ are weighting hyper-parameters that we empirically set to 1 and 0.1, respectively.

Table 1: Basic statistics of the EventDC benchmark.

| Dataset | Color Camera | Depth Sensor | Event Camera | Train | Test | Resolution |
|---------|-------------|--------------|--------------|-------|------|-----------|
| EventDC-Real | FLIR BFS-U3-31S4C | Ouster OS1-128 LiDAR | DAVIS346 | 14,845 | 1,000 | $320 \times 256$ |
| EventDC-SemiSyn | PointGrey Flea2 | Velodyne HDL-64E LiDAR | - | 7,094 | 2,213 | $1216 \times 256$ |
| EventDC-FullSyn | - | - | - | 21,000 | 500 | $512 \times 256$ |

| Color image | Sparse depth | Event stream | GT depth |

Figure 3: Visualizations of the proposed EventDC benchmark: EventDC-Real/SemiSyn/FullSyn.

## 4 EventDC Benchmark

**Motivation.** Traditional depth completion datasets [14, 42, 65, 44] rely on the fusion of color images and sparse depth maps to predict dense depth. However, this approach suffers in highly dynamic environments especially when dealing with fast ego-motion and object motion. This is due to unreliable low-frame-rate RGB images and sparse depth data from motion blur and sampling inconsistencies. Event cameras with the capability to capture high temporal resolution and sensitivity to rapid movements [10] provide an ideal solution to overcome these limitations. By asynchronously recording minute brightness variations, event cameras can offer accurate depth information in dynamic scenarios where conventional RGB-D sensors fail. In light of these characteristics, we propose an event-based depth completion benchmark that leverages the unique advantages of event data to address the challenges of depth completion in dynamic environments.

**Data Collection.** Tab. 1 provides an overview of the sensors used in the datasets with their respective specifications. *EventDC-Real* is a real-world dataset in which color images and event frames are captured using the FLIR BFS-U3-31S4C camera and the DAVIS346 sensor, respectively. The ground truth (GT) depth is acquired from a 128-line Ouster LiDAR, and the sparse depth is derived from its 16 sub-lines. *EventDC-SemiSyn* is a semi-synthetic dataset based on KITTI [14]. The sparse depth and GT depth come from the raw data of KITTI. For the color images, we apply radial motion blur by progressively scaling and transforming the image around its center to simulate a motion blur effect with adjustable strength and step count. Additionally, VID2E [12] is used to generate the event data with frames captured within 15 ms before and after the current timestamp. *EventDC-FullSyn* is a fully synthetic dataset generated using the CARLA simulator [6]. The color images are processed similarly with radial motion blur. Finally, to facilitate model training, the resolution of all datasets has been cropped to multiples of 32. Fig. 3 presents visual examples from these three datasets.

## 5 Experiment

**Metric and Implementation Detail.** Following previous depth completion methods [14, 46, 65, 26], we adopt RMSE (mm), MAE (mm), REL, and threshold accuracy $\delta$ (%) as evaluation metrics. Refer to the appendix for their full definitions. We implement EventDC using the PyTorch framework and conduct training on two NVIDIA RTX 4090 GPUs using the Distributed Data Parallel strategy for efficiency. Optimization is performed with the AdamW optimizer [31] in conjunction with the OneCycle learning rate policy [43]. The training process begins with a warm-up stage that linearly increases the learning rate from 0.00002 to 0.001 over the first 10% of iterations. Subsequently, a cosine annealing schedule gradually decays the learning rate to a final value of 0.0002. The batch size is set to 2 per GPU. In addition, to further enhance model performance, we employ a set of data augmentation strategies [46, 29], including random horizontal flip, rotation, cropping, and color jitter.

Table 2: Quantitative depth completion comparisons on the EventDC-Real dataset.

| Method | RMSE ↓ | MAE ↓ | REL ↓ | $\delta_{1.05}$ ↑ | $\delta_{1.10}$ ↑ | $\delta_{1.15}$ ↑ | Venue |
|---|---|---|---|---|---|---|---|
| CSPN [2] | 858.5 | 284.6 | 0.0386 | 90.0 | 94.4 | 96.0 | ECCV 2018 |
| S2D [33] | 984.1 | 410.8 | 0.0565 | 82.1 | 90.3 | 93.4 | ICRA 2018 |
| FusionNet [49] | 658.1 | 262.4 | 0.0384 | 87.6 | 93.9 | 96.0 | MVA 2019 |
| RigNet [61] | 685.4 | 234.6 | 0.0336 | 87.5 | 92.3 | 95.4 | ECCV 2022 |
| DySPN [27] | 700.1 | 223.7 | 0.0285 | 91.3 | 95.1 | 96.6 | AAAI 2022 |
| Prompting [38] | 670.7 | 205.1 | 0.0252 | 92.1 | 95.7 | 97.0 | CVPR 2024 |
| OGNI-DC [78] | 709.7 | 231.1 | 0.0294 | 90.9 | 94.8 | 96.4 | ECCV 2024 |
| SigNet [66] | 906.4 | 348.1 | 0.0345 | 83.5 | 90.6 | 93.2 | CVPR 2025 |
| LPNet [55] | 911.2 | 389.0 | 0.0472 | 83.6 | 90.4 | 93.3 | arXiv 2025 |
| **EventDC (our)** | **574.0** | **179.0** | **0.0242** | **92.9** | **96.3** | **97.5** | - |
| *Improvement* ↑ | *84.1* | *26.1* | *0.0010* | *0.8* | *0.6* | *0.5* | - |

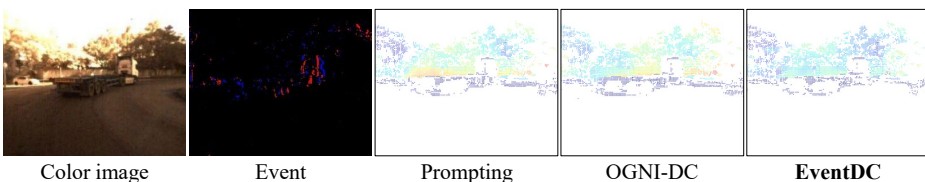

| Color image | Event | Prompting | OGNI-DC | **EventDC** |
|---|---|---|---|---|

Figure 4: Depth error comparisons on EventDC-Real. Warmer color indicates higher error.

## 5.1 Comparisons with State-of-the-arts

In this section, we compare our EventDC with well-known methods: CSPN [2], S2D [33], FusionNet [49], RigNet [61], DySPN [27], Prompting [38], OGNI-DC [78], SigNet [66], and LPNet [55]. For a fair comparison, we retrain all methods from scratch on the proposed benchmark. Note that BPNet [47], TPVD [65], and DMD³C [26] are excluded from the comparison. This is because they require additional camera parameters during training which are not available in our settings.

**EventDC-Real.** We first evaluate the proposed EventDC on EventDC-Real, a real-world dataset collected using various devices such as handheld sensors and robotic platforms. The numerical results are summarized in Tab 2. Our EventDC achieves the overall lowest errors while maintaining the highest accuracy across the board. For example, it outperforms the second-best method by 84.1 mm in RMSE, 26.1 mm in MAE, 0.001 in REL, and 0.8 points in $\delta_{1.05}$. Compared to post-refinement methods such as CSPN [2] and DySPN [27], our EventDC without any post-processing consistently achieves better performance. Even when compared to the large-scale depth foundation model Prompting [38], our approach achieves superior results with significantly fewer model parameters. Fig. 4 presents the comparisons of depth error. It clearly shows that our EventDC produces more accurate depth results especially around moving objects.

**EventDC-SemiSyn.** To further validate the effectiveness of EventDC, we evaluate it on EventDC-SemiSyn, a semi-synthetic dataset comprising synthetically generated event frames and color images rendered under highly dynamic conditions. As reported in Tab. 3, our EventDC continues to deliver outstanding results across all evaluation metrics. On average, it outperforms recent methods: Prompting [38], OGNI-DC [78], and LPNet [55] by 18.9%, 30.8%, and 27.7% in RMSE, MAE, and REL, respectively, and by 3.9, 1.6, and 0.9 percentage points in $\delta_{1.05}$, $\delta_{1.10}$, and $\delta_{1.15}$, respectively. As illustrated in Fig. 5, our EventDC effectively reconstructs accurate depth details and structural consistency even under highly dynamic scenes.

**EventDC-FullSyn.** Apart from the real and semi-synthetic settings, we also validate EventDC on the fully synthetic dataset, EventDC-FullSyn, to further assess its generalization capability under diverse scenarios. As shown in Tab.4, our EventDC consistently outperforms all competing approaches by large margins. For example, it surpasses the second-best approach by 53.5 mm in RMSE and 19.4 mm in MAE. In addition, it achieves a 13.0% improvement in REL compared to the foundation model-based Prompting [38]. These results demonstrate the robustness of our EventDC in reducing both absolute and relative errors. Fig. 6 shows that our EventDC yields more refined details and sharper object boundaries than others, which highlight its effectiveness in fully synthetic scenarios.

Table 3: Quantitative comparisons on the EventDC-SemiSyn dataset.

| Method | RMSE↓ | MAE↓ | REL↓ | $\delta_{1.05}$ ↑ | $\delta_{1.10}$ ↑ | $\delta_{1.15}$ ↑ |
|---|---|---|---|---|---|---|
| CSPN [2] | 989.8 | 262.8 | 0.0189 | 94.6 | 97.2 | 98.1 |
| S2D [33] | 1097.3 | 366.4 | 0.0237 | 91.0 | 96.4 | 97.9 |
| FusionNet [49] | 877.6 | 333.1 | 0.0258 | 92.6 | 98.2 | 98.8 |
| RigNet [61] | 858.2 | 216.4 | 0.0156 | 95.1 | 97.8 | 98.1 |
| DySPN [27] | 897.7 | 207.5 | 0.0149 | 95.9 | 97.8 | 98.6 |
| Prompting [38] | 873.9 | 291.1 | 0.0198 | 92.6 | 97.1 | 98.4 |
| OGNI-DC [78] | 832.0 | 210.5 | 0.0143 | 95.7 | 98.0 | 98.7 |
| SigNet [66] | 1065.4 | 321.3 | 0.0226 | 91.1 | 97.0 | 98.1 |
| LPNet [55] | 1283.4 | 416.3 | 0.0242 | 90.0 | 95.3 | 97.2 |
| **EventDC (ours)** | **778.8** | **196.2** | **0.0134** | **96.7** | **98.4** | **99.0** |
| *Improvement* ↑ | *53.2* | *11.3* | *0.0009* | *0.8* | *0.2* | *0.2* |

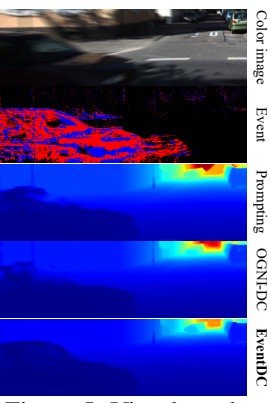

Figure 5: Visual results.

Table 4: Quantitative comparisons on the EventDC-FullSyn dataset.

| Method | RMSE↓ | MAE↓ | REL↓ | $\delta_{1.05}$ ↑ | $\delta_{1.10}$ ↑ | $\delta_{1.15}$ ↑ |
|---|---|---|---|---|---|---|
| CSPN [2] | 864.9 | 399.5 | 0.1193 | 62.7 | 80.9 | 87.6 |
| S2D [33] | 899.0 | 376.2 | 0.1243 | 69.8 | 83.8 | 89.1 |
| FusionNet [49] | 670.6 | 230.9 | 0.0931 | 77.3 | 86.6 | 90.4 |
| RigNet [61] | 723.4 | 166.3 | 0.0578 | 81.1 | 91.6 | 92.8 |
| DySPN [27] | 679.8 | 165.6 | 0.0646 | 87.2 | 92.6 | 94.6 |
| Prompting [38] | 709.7 | 180.9 | 0.0538 | 90.7 | 93.8 | 95.3 |
| OGNI-DC [78] | 673.7 | 162.5 | 0.0578 | 87.8 | 93.0 | 95.1 |
| SigNet [66] | 904.5 | 349.2 | 0.0902 | 76.3 | 84.1 | 90.3 |
| LPNet [55] | 920.2 | 357.3 | 0.0943 | 75.1 | 85.9 | 90.3 |
| **EventDC (ours)** | **620.2** | **143.1** | **0.0468** | **92.1** | **95.5** | **96.8** |
| *Improvement* ↑ | *53.5* | *19.4* | *0.0070* | *1.4* | *1.7* | *1.5* |

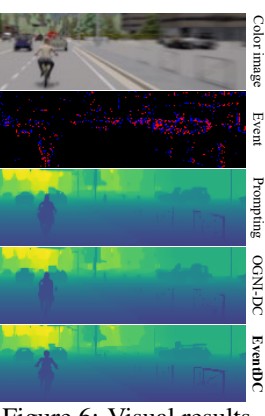

Figure 6: Visual results.

**Complexity Analysis.** Tab. 5 presents the complexity comparisons between our EventDC and other competing methods in terms of model parameters (Param.), memory consumption (Memo.), and inference time. Our EventDC not only achieves outstanding performance, but also maintains competitive efficiency. In particular, compared to the second-best method Prompting [38], our EventDC achieves a significantly lower RMSE by 96.7 mm with only about one-eighth the number of parameters and one-third the memory.

Table 5: Complexity on EventDC-Real.

| Method | Param. (M) ↓ | Memo. (GB) ↓ | Time (ms) ↓ | RMSE (mm) ↓ |
|---|---|---|---|---|
| DySPN [27] | **26.3** | **0.9** | **9.8** | 700.1 |
| RigNet [61] | 65.2 | 2.3 | 26.5 | 685.4 |
| Prompting [38] | 326.9 | 4.1 | 39.5 | 670.7 |
| OGNI-DC [78] | 84.4 | 3.7 | 314.1 | 709.7 |
| LPNet [55] | 29.6 | 1.1 | 18.4 | 911.2 |
| **EventDC (our)** | 43.2 | 1.5 | 41.5 | **574.0** |

## 5.2 Ablation Studies

Tab. 6 summarizes the ablation results on EventDC-Real. EventDC-i serves as a UNet-style baseline that takes only sparse depth as input and employs additive skip connections.

**(1)** EventDC-ii further develops this approach by utilizing RGB images and integrating RGB-D features through additive fusion. Although depth input is sparse, RGB offers rich structural and semantic details. This leads to a significant decrease in error and substantial gains in accuracy. For example, the RMSE is reduced by 41.3 mm and the MAE by 32.1 mm. EventDC-iii enhances support for event streams, which are advantageous because of their fine temporal detail and motion sensitivity. Consequently, this makes them very effective in dynamic environments where they supplement depth data. EventDC-iv combines all three modalities to give consistent improvements across all evaluation metrics. Specifically, it surpasses the baseline by 12.2%, 20.7%, and 8.5% in RMSE, MAE, and REL, respectively. It concurrently improves $\delta_{1.05}$, $\delta_{1.10}$, and $\delta_{1.15}$ by 1.0, 0.3, and 0.2 percentage points. These results underscore the effectiveness of multi-modal fusion, where the integration of complementary modalities enables more accurate and complete depth reconstruction.

Table 6: Ablations on EventDC-Real. DConv/Enc/Dec: dynamic convolution/encoder/decoder.

| EventDC | Modality | | | DConv | | EMA | LDF | RMSE | MAE | REL | $\delta_{1.05}$ | $\delta_{1.10}$ | $\delta_{1.15}$ |
|---|---|---|---|---|---|---|---|---|---|---|---|---|---|
| | Depth | RGB | Event | Enc | Dec | Enc | Dec | | | | | | |
| i | ✓ | | | | | | | 727.2 | 283.2 | 0.0341 | 89.3 | 94.4 | 96.2 |
| ii | ✓ | ✓ | | | | | | 685.9 | 251.1 | 0.0328 | 89.8 | 94.5 | 96.2 |
| iii | ✓ | | ✓ | | | | | 696.0 | 244.3 | 0.0314 | 90.0 | 94.5 | 96.3 |
| iv | ✓ | ✓ | ✓ | | | | | 638.3 | 224.7 | 0.0312 | 90.3 | 94.7 | 96.4 |
| v | ✓ | ✓ | ✓ | ✓ | | | | 628.8 | 219.5 | 0.0292 | 91.0 | 95.0 | 96.7 |
| vi | ✓ | ✓ | ✓ | | | ✓ | | 602.8 | 196.1 | 0.0276 | 91.7 | 95.6 | 97.1 |
| vii | ✓ | ✓ | ✓ | | ✓ | | | 630.6 | 219.8 | 0.0295 | 91.0 | 94.8 | 96.5 |
| viii | ✓ | ✓ | ✓ | | | | ✓ | 605.4 | 198.3 | 0.0279 | 91.5 | 95.6 | 97.1 |
| ix | ✓ | ✓ | ✓ | | | ✓ | ✓ | 574.0 | 179.0 | 0.0242 | 92.9 | 96.3 | 97.5 |

(a) Histogram comparison

(b) Results around moving objects

Figure 7: Statistical and visual comparative analyses of the proposed EMA and LDF modules.

**(2)** EventDC-v to EventDC-ix conduct ablation studies to examine the impact of dynamic convolution (DConv), EMA, and LDF in the encoder (Enc) and decoder (Dec) stages. Specifically, the introduction of DConv in EventDC-v brings notable benefits. Furthermore, EventDC-vi with EMA further reduces RMSE by 26 mm. These results validate the efficacy of our event-based adaptive alignment strategy. Fig. 7(a) compares the distributions of RGB-D features with and without EMA. EMA works as intended in promoting better alignment between the two modalities with more consistent feature representations. Similarly, EventDC-viii with LDF outperforms EventDC-vii with DConv by 25.2 mm. This demonstrates its superior ability to recover fine-grained local depth which is further evident in Fig. 7(b). Finally, EventDC-ix which integrates both EMA and LDF modules achieves the best overall performance. It reduces RMSE by 16.0% (from 683.3 mm) and MAE by 20.3% (from 224.7 mm). In summary, each component contributes positively to the overall performance gains.

## 6 Conclusion

We propose EventDC in this work. Our EventDC is the first depth completion framework that tackles the challenges of dynamic scenes by harnessing the unique strengths of event data. To mitigate the adverse effects of fast ego-motion and object motion, our EventDC incorporates two event-driven modules: event-modulated alignment and local depth filtering. These modules, supported by two dedicated loss constraints, address global misalignment and local depth inaccuracies, respectively. To further support research in this area, we construct the first benchmark for event-based depth completion comprising one real-world and two synthetic datasets. Extensive experiments demonstrate the effectiveness of our EventDC and its superior performance in challenging dynamic environments.

**Limitation and Broader Impact.** Despite achieving promising results in dynamic scenes, our EventDC relies on high-quality event data and precise sensor alignment that may not be easily attainable in all real-world settings. The EMA and LDF modules introduce additional computational costs, potentially limiting deployment on resource-constrained devices. Moreover, the scale and diversity of our real-world dataset are limited, and future work is needed to evaluate generalization across more diverse environments and motion patterns. Despite these limitations, our EventDC offers a step forward in robust depth perception under motion blur and rapid dynamics with potential applications in autonomous driving, robotics, AR/VR, *etc*. By introducing a dedicated benchmark, we aim to promote research in event-based depth completion. As with all perceptual systems, responsible deployment requires attention to reliability, fairness, and safety in complex real-world conditions.

## Acknowledgments

This research/project is supported by the National Research Foundation, Singapore, under its NRF-Investigatorship Programme (Award ID. NRF-NRFI09-0008) and the Tier 1 grant T1-251RES2305 from the Singapore Ministry of Education.

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

# A  Metrics.

We adopt the Root Mean Squared Error (RMSE), Mean Absolute Error (MAE), Mean Relative Error (REL), and threshold accuracy $\delta_\theta$ as evaluation metrics, where $\theta$ is set to 1.05, 1.10, and 1.15. The definitions of these metrics are shown in Tab. 7.

Table 7: Definition of evaluation metrics.

Given the predicted depth $\mathbf{D}$ and GT depth $\mathbf{Z}$ with $n$ valid pixels:

| | | | |
|---|---|---|---|
| – RMSE: | $\sqrt{\frac{1}{n}\sum(\mathbf{D}-\mathbf{Z})^2}$ | – MAE: | $\frac{1}{n}\sum|\mathbf{D}-\mathbf{Z}|$ |
| – REL: | $\frac{1}{n}\sum|\mathbf{D}-\mathbf{Z}|/\mathbf{Z}$ | – $\delta_\theta$: | $\frac{q}{n},\ q:\max\left(\frac{\mathbf{D}}{\mathbf{Z}},\frac{\mathbf{Z}}{\mathbf{D}}\right)<\theta$ |

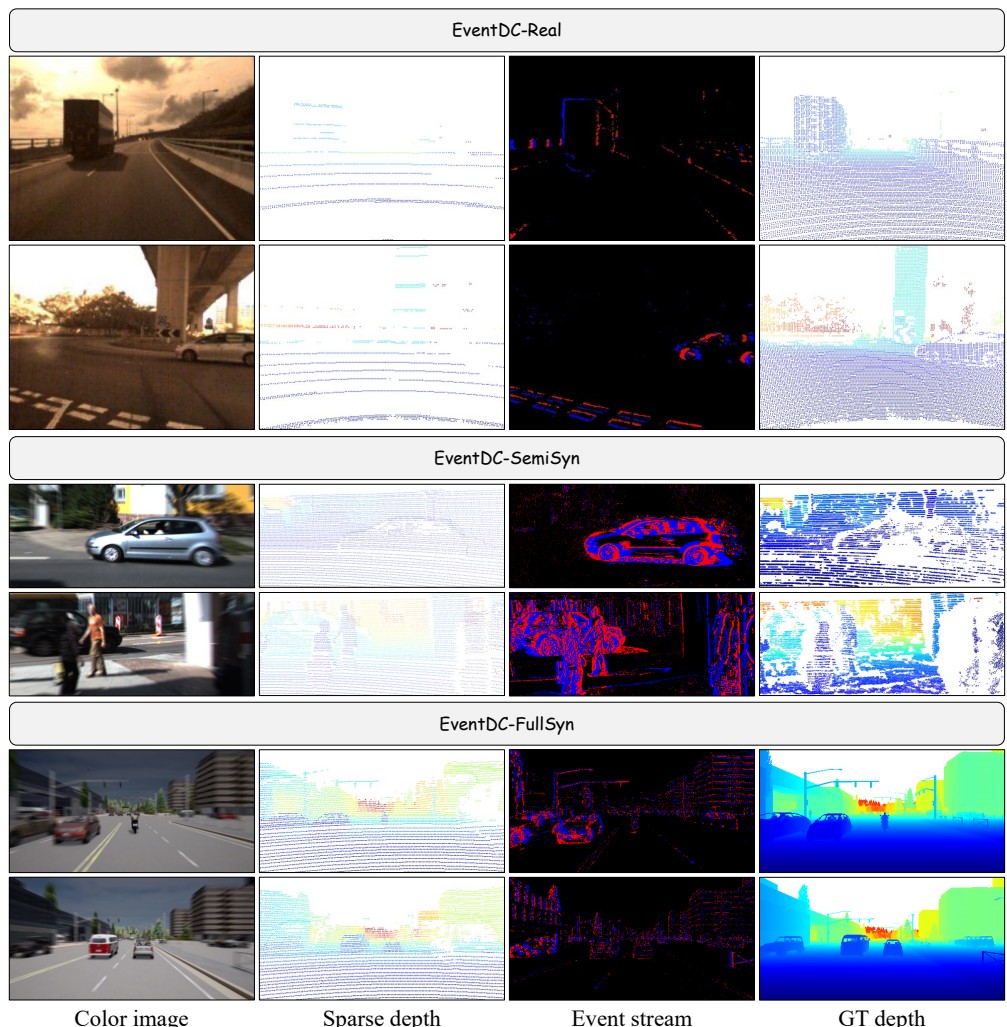

Figure 8: More visual examples of the proposed event-based depth completion benchmark.

# B  More Visualizations

Fig. 8 presents some RGB-D-Event examples from our event-based depth completion benchmark, showcasing its high quality and strong cross-modal consistency, as well as the close correlation among the RGB, depth, and event modalities. Figs. 9, 10 and 11 show visual comparisons on the EventDC-Real, EventDC-SemiSyn, and EventDC-FullSyn datasets. These results further validate that our approach effectively improves depth predictions through the event-driven module designs.

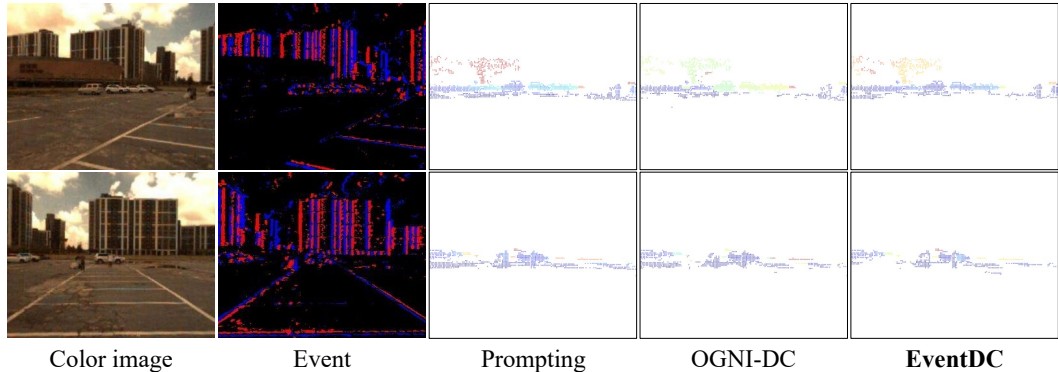

| Color image | Event | Prompting | OGNI-DC | **EventDC** |

Figure 9: More depth error comparisons on EventDC-Real. Warmer colors indicate higher errors.

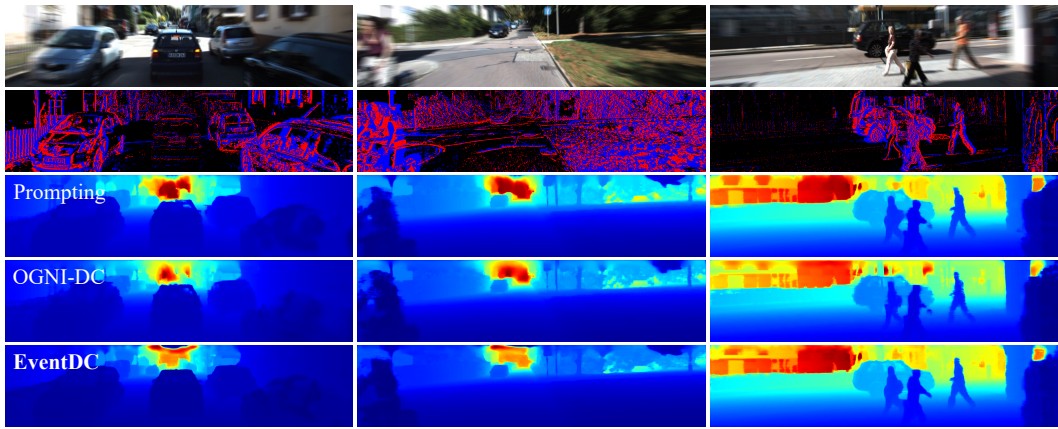

Figure 10: More depth visualization comparisons on the proposed EventDC-SemiSyn dataset.

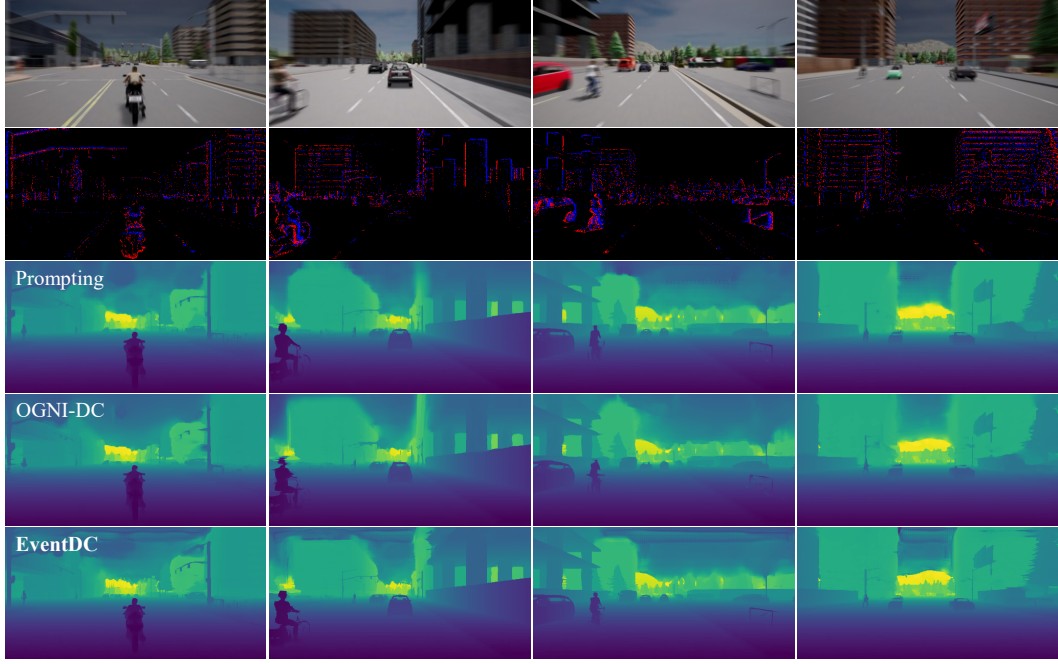

Figure 11: More depth visualization comparisons on the proposed EventDC-FullSyn dataset.

