# OpenReview forum: "Event-Driven Dynamic Scene Depth Completion"
_NeurIPS.cc/2025/Conference — NeurIPS 2025 poster_

### Official Review · Reviewer_G3qD · 2025-06-27

**Clarity:** 4
**Significance:** 4
**Originality:** 3
**Rating:** 6
**Confidence:** 5

**Summary:**

This paper proposes EventDC, the first event-driven depth completion framework tailored for dynamic scenes. It addresses the common challenges of RGB-LiDAR misalignment and depth degradation caused by fast ego-motion and object motion, which conventional sensors and fusion approaches struggle to handle. The method introduces two novel modules:

1. Event-Modulated Alignment (EMA): leverages event features to predict spatial offsets for RGB and LiDAR features, improving global alignment and multi-modal fusion during encoding.

2. Local Depth Filtering (LDF): focuses on decoder-side refinement near moving objects, using event-informed motion masks and adaptive depth filtering.

In addition, the paper presents the first benchmark for event-based depth completion, including real-world (EventDC-Real), semi-synthetic (EventDC-SemiSyn), and synthetic (EventDC-FullSyn) datasets. Extensive experiments show consistent SOTA performance across all settings, with detailed ablation studies and complexity analysis.

**Questions:**

1. Use of the benchmark: Do you plan to release the simulation or preprocessing tools used to create EventDC-SemiSyn and EventDC-FullSyn? This could encourage broader community engagement.

2. Failure cases: If possible, could you describe typical scenarios where EventDC might underperform (e.g., repetitive textures or extremely low motion regions)? Even a qualitative description would enhance the paper’s completeness.

**Ethical Concerns:**

["NO or VERY MINOR ethics concerns only"]

**Final Justification:**

I agree with Reviewer 1o7Y. This paper makes meaningful contributions by tackling the underexplored problem of event-based depth completion, effectively filling a gap in the field. I am willing to raise my score.

**Limitations:**

Yes.

**Quality:**

4

**Strengths And Weaknesses:**

Strengths:

1. Interesting problem and new benchmark datasets: This paper addresses a real yet underexplored scenario, i.e., depth completion in highly dynamic scenes, where existing RGB-D systems underperform. Furthermore, It introduces a three-part benchmark (real/semi-synthetic/full synthetic), which is a notable contribution to the field, likely to promote further research in event-based depth learning.

2. Well-motivated design: The EMA and LDF modules introduce event-driven dynamic convolution for both global alignment and local refinement, exploiting events’ high temporal resolution effectively.

3. Strong performance: EventDC outperforms 9+ strong baselines on three datasets with consistent improvements in many metrics without post-processing or additional priors.

4. Comprehensive experiments: Detailed studies isolate the contribution of each modality (RGB, event, sparse depth), component (EMA/LDF), and design choices (dynamic convolution), which strengthens the claims. Comprehensive experiments show the superiority of the proposed method.

Minor Weaknesses:

While the paper is overall strong and well-executed, a few aspects could benefit from additional clarification or discussion:

1. Dependence on high-quality events: As acknowledged, the method assumes reliable event streams and well-synchronized RGB-D-event input. A brief discussion on robustness to sensor noise or misalignment would improve practical insights.

2. Benchmark diversity: While the EventDC benchmark is a valuable contribution, especially with a real-world component, its coverage of diverse environmental conditions (e.g., outdoor lighting, long-range scenes) could be discussed to help readers understand the scope and future extension possibilities.

None of these points undermine the core contributions, and they can be addressed through minor clarifications.

---

> ### Author Rebuttal · Authors · 2025-07-28
>
> **Minor Weakness1:**
>
> Although our method assumes reasonably synchronized input, it is explicitly designed to handle the challenges of imperfect RGB-D alignment in high-speed or dynamic scenes. In such cases, event data which is naturally asynchronous and high-temporal-resolution, plays a key role in correcting spatial misalignments and compensating for motion blur or sparse/degraded depth near dynamic objects.
>
> To this end, we introduce event-guided deformable convolution, where the event stream dynamically generates the offset fields and modulation weights. This allows the model to adaptively sample features from more accurate spatial positions, even in the presence of RGB-D misalignment. In other words, instead of requiring perfectly aligned input, our method leverages the asynchronous nature of events to mitigate such issues.
>
> We will add a discussion of this point in the revision, to clarify that our approach is particularly beneficial in non-ideal, high-motion scenarios, and offering future directions such as noise-aware offset estimation or sensor calibration refinement for further robustness. Thank you for the insightful suggestion.
>
> **Minor Weakness2:**
>
> We would like to emphasize that the EventDC benchmark already includes a diverse set of environmental conditions to facilitate comprehensive evaluation. Specifically:
>
> *1. Scene diversity:* Our dataset covers both indoor and outdoor scenarios, ranging from confined corridors to large open areas.
>
> *2. Lighting variation:* We include sequences captured in daytime, nighttime, and low-light conditions, making the benchmark suitable for evaluating models under challenging illumination.
>
> *3. Depth range:* The effective depth spans from approximately 0.1 meters to 100 meters, thus covering both close-range tasks (e.g., indoor navigation or robotic grasping) and long-range perception (e.g., outdoor driving or surveillance).
>
> **Question1:**
>
> Yes, we will release the proposed datasets upon paper acceptance. In addition, we are glad to share the corresponding preprocessing tools to further support and benefit the research community.
>
> **Question2:**
>
> We agree that discussing failure cases can provide a more complete understanding of our method's limitations. EventDC, while effective in high-motion scenarios, may underperform in the following typical cases:
>
> *1. Extremely low-motion or static regions:* Since event data is inherently triggered by intensity changes, regions with little or no motion (e.g., static background or slowly moving objects) generate very few events. This limits the contribution of event-guided modules in such areas, making the model rely primarily on RGB and sparse depth.
>
> *2. Repetitive or low-texture areas:* In scenes with repetitive patterns or low texture (e.g., blank walls or road surfaces), events may be ambiguous or noisy, leading to inaccurate offset estimation in deformable convolutions.
>
> *3. Sensor artifacts or miscalibration:* Although our method is designed to handle RGB-D misalignment, large-scale calibration errors or noisy depth input may still affect performance, especially in complex outdoor scenes.
>
> *Thank you again for these valuable comments. We will carefully consider incorporating these discussions in the revision.*

---

> > ### Comment · Reviewer_G3qD · 2025-08-01
> >
> > I appreciate the authors' responses, as my concerns have been adequately addressed. After carefully reading the other reviewers' comments and the authors’ rebuttals, I maintain my recommendation for acceptance. This is a solid and interesting piece of work. In highly dynamic scenes, event cameras offer inherent advantages, and this paper effectively leverages them. More importantly, it introduces the first event-driven depth completion benchmark, which is a valuable contribution to the community.
> >
> > While EventDC adopts commonly used deformable convolutions, it thoughtfully redesigns them from both data-specific and task-specific perspectives, with clear and well-motivated reasoning. Additionally, since this is the first work to propose an event-based depth completion benchmark, the authors had to retrain all competing methods on the three datasets, which represents a substantial amount of work.
> >
> > I believe all reviewers have invested considerable time in evaluating this paper, and their feedback has helped improve it. We should continue to encourage and support its further refinement.
> >
> > Overall, I recommend acceptance. I believe this work is insightful and high-quality, which can make significant contributions to the depth completion community.

---

> > > ### Author Response · Authors · 2025-08-01
> > > **Release of datasets and processing tools**
> > >
> > > Thanks for the support! The proposed three datasets have been packaged and are scheduled for release in two months, together with the corresponding processing tools.

---

### Official Review · Reviewer_C3TZ · 2025-07-03

**Clarity:** 4
**Significance:** 3
**Originality:** 3
**Rating:** 4
**Confidence:** 4

**Summary:**

The paper introduces a new event-driven depth completion framework, named EventDC, which is designed to address the challenges of dynamic environments. The model mainly consists of two modules, Event-Modulated Alignment (EMA) and Local Depth Filtering (LDF), to mitigate the misalignment caused by ego-motion and local depth inaccuracies due to object motion. Two dedicated loss constraints are used for supporting training. Besieds, it builds an event-based benchmark for depth completion.

**Questions:**

As listed in the "Weaknesses". I would consider raise the score if the rebuttal is satisfying.

**Ethical Concerns:**

["NO or VERY MINOR ethics concerns only"]

**Final Justification:**

Considering the rebuttal and discussions from other reviewers, I think the paper proposes a good work for event-based vision. The benchmark and the topic is good. I raise my rating after careful consideration.

**Limitations:**

None.

**Quality:**

3

**Strengths And Weaknesses:**

Strengths:
1. The paper is well-written and formatted.
2. In this paper's setting, the results seems outperform other method of depth completion.
3. The EventDC benchmark is a good baseline for future researchs.
Weakness:
1. In real world, if there are rich and complicated motions in the environment, not only be the color image not reliable, the sparse depth is not reliable neither. In this situation, how to and why use event streams to fix or complete the unreliable depth map?
2. The model architecture lacks novelty. In the EventDC, the EMA and LDF all consists of basic DCN blocks, which is a commonly-used module.
3. Another key concern is the fairness of the comparison experiments. While in the motion-rich environments, the image data are designed or set to possessing blur artifacts. It may not fair for image-based methods. Are there futher experiments for demonstration?

---

> ### Author Rebuttal · Authors · 2025-07-28
>
> **To Weakness4:**
>
> Indeed, in real-world dynamic scenes with fast and complex motion, both RGB frames and sparse depth may become unreliable due to motion blur, temporal misalignment, or sensor noise. This is precisely the motivation behind leveraging event streams, which provide asynchronous, and high-temporal-resolution measurements that are inherently robust to fast motion and extreme lighting conditions. Event data captures pixel-level brightness changes at microsecond latency, making it resilient to motion blur and misalignment that commonly affect RGB and depth sensors.
>
> In our framework, events are used to guide the spatial sampling of features through event-driven deformable convolutions. The offsets and modulation weights in these convolutions are derived from event features. For example, as illustrated in Figure 2, the event-centric EMA module adjusts the pixel distributions of RGB and depth features using event information. This enables more precise RGB-D alignment (see Figure 7(a)). The design remains effective even when RGB and depth inputs become unreliable. This is due to the microsecond-level temporal resolution and high motion sensitivity of event streams in dynamic environments.
>
> Although RGB and sparse depth provide appearance and geometric priors, events offer fine-grained motion cues that help localize trustworthy structures and correct ambiguous or missing depth information in challenging scenarios. This synergy enables our model to better recover complete and reliable depth maps under conditions where traditional sensors struggle.
>
> Finally, we would like to further clarify our current focus:
>
> Our work focuses on highly dynamic scenes where RGB-D misalignment and local depth degradation commonly occur near moving objects or during fast camera motion. Although RGB and sparse depth are generally informative in most parts of the scene, we explicitly target regions where they become less reliable due to motion blur or latency mismatch. Thus, the proposed setting does not assume that both RGB and sparse depth are globally unreliable. Instead, it accounts for their reliability being spatially and temporally variant, which is common in real-world dynamic scenarios. We thank the reviewer for this insightful question, and will add the discussion into the final paper.
>
>
> **To Weakness5:**
>
> Thank you for raising this important concern. It is true that DCN is a widely used module. However, the contribution of this work does not lie in merely using DCN. Instead, it lies in how DCN is adapted and leveraged specifically for event-guided depth completion in highly dynamic scenes.
>
> **First**, both the EMA and LDF modules are carefully designed to tackle the unique challenges posed by RGB-D misalignment and unreliable depth around moving objects in dynamic environments. In particular:
>
> (a) The EMA module uses event features to dynamically generate DCN offsets and weights. This enables spatially adaptive feature alignment across RGB and depth inputs. To the best of our knowledge, this is one of the first methods proposed to use events to guide deformable alignment for depth completion in dynamic scenes.
>
> (b) The LDF module further refines depth estimations around moving objects by learning motion-aware masks from events, which is crucial in scenes with locally degraded or missing depth.
>
> **Second**, our integration with event data is notably different although DCN is a standard building block. This includes our design of the supervision and training strategy. For example, how the offset is driven by event features. In contrast to conventional DCN usage, which typically operates in a generic spatial context, our formulation is event-centric. It uses the microsecond-level temporal cues provided by event streams.
>
> **Finally**, the novelty of our work is not limited to architecture, but also includes the problem setting, i.e., event-guided depth completion in dynamic RGB-D scenes, and the benchmark contribution (EventDC dataset), which enables a more rigorous study of this underexplored problem.
>
> We hope this explanation clarifies the novelty and motivation of our design. We will revise the manuscript to better highlight these aspects. Thank you again for the constructive feedback!
>
>
> **To Weakness6:**
>
> Thank you for raising this important concern. We would like to clarify that our comparison setting is **fair** and **rigorous**. All compared methods fall under the category of depth completion, which take both color images and sparse depth as inputs, **not image alone**. Similarly, our EventDC method also takes color images and sparse depth as inputs, with the addition of event streams.
>
> Our EventDC benchmark comprises three distinct subsets, i.e., *EventDC-Real*: Real-world RGB-D-event recordings *without any synthetic motion blur*; *EventDC-SemiSyn* and *EventDC-FullSyn*: Semi-synthetic and fully synthetic datasets *with simulated motion blur* to emulate high-dynamic scenes.
>
> This setup ensures that our evaluation covers a wide spectrum of motion conditions, from **real-world clean input** to **extreme motion-degraded cases**. In particular, EventDC-Real **does not contain any artificially induced blur**, and yet our method still outperforms image-based baselines. This provides strong evidence that our model is effective even without relying on synthetic motion artifacts.
>
> For EventDC-SemiSyn and EventDC-FullSyn, motion blur is intentionally added to simulate fast motion scenarios. These scenarios are the core focus of our work. The goal is to benchmark the robustness of RGB-D and event-based methods under such conditions. This is not intended to bias the results against image-based methods, but to fairly reflect real-world challenges where event streams offer the greatest benefit.
>
> Furthermore, all methods are evaluated on the exact same inputs to ensure fairness in the comparison. The advantage of our model under motion blur conditions arises from its ability to leverage microsecond-resolution event data, which is inherently lacking in image-based models.
>
> Additionally, to further support the fairness of our evaluation, we provide the following results on a version of EventDC-FullSyn where no motion blur is applied to color images:
>
> *Table R4: Results on EventDC-FullSyn without motion blur.*
> | Method | RMSE | MAE |
> |---|---|---|
> | CSPN     | 840.6 | 391.8 |
> |OGNI-DC | 657.5 | 147.6 |
> |EventDC  | 608.3 | 139.4 |
>
> *We appreciate the reviewer’s comment in helping us to improve the clarity and completeness of our work. We will add these results in the revision.*

---

> > ### Author Response · Authors · 2025-08-07
> >
> > Dear our respectable Reviewer C3TZ,
> >
> > Thank you again for your valuable feedback, and we sincerely appreciate your positive ratings on our paper’s Quality, Clarity, Significance, and Originality! We have now addressed the concerns raised by the other three reviewers, and *your opinion is also very important to us*. We would like to know whether our reply has satisfactorily addressed your comments. As the author–reviewer discussion period will end in about *48 hours*, please let us know if you have any further questions. We will do our best to address them, thanks!
> >
> > Best regards,
> >
> > Authors

---

> > ### Comment · Reviewer_C3TZ · 2025-08-07
> >
> > The authors have partially resolved my concerns. Though the mode architecture is not novel enough, the benchmark and the experiments are useful for event-based vision. The details in the reply should applied to the main paper in the later revision. I raise my rating considering the rebuttal and discussions from other reviewers.

---

> > > ### Author Response · Authors · 2025-08-07
> > >
> > > We appreciate your kind support! We will add those details we discussed in our revision.

---

### Official Review · Reviewer_1o7Y · 2025-07-03

**Clarity:** 3
**Significance:** 3
**Originality:** 3
**Rating:** 4
**Confidence:** 4

**Summary:**

This paper proposes an EventDC model that leverages RGB, LiDAR, and event images to generate dense depth maps. Two key components—event-modulated alignment and local depth filtering—are designed based on deformable convolution. Meanwhile, three datasets are constructed, including a real-world dataset, a semi-synthetic dataset, and a fully synthetic dataset. Experiments on these datasets demonstrate the effectiveness of the proposed method.

**Questions:**

1. In Figure 2, should the Event stream E be positioned in the middle while the sparse depth S is at the bottom?

2. It is suggested to unify the notation of the offset in equations (1), (3a), and (3b).

3. Why are the weight designs in the deformable convolution different between the encoder and decoder? What would happen if they used the same design?

4. It is expected that EventDC improves depth estimation around moving objects, but in Figure 4, it appears that improvements are more pronounced in the distant background, which might be less dynamic. Could you explain why? It would be helpful to see more results on dynamic objects.

5. How does the model perform without the event stream but with the designed deformable convolutions?

**Ethical Concerns:**

["NO or VERY MINOR ethics concerns only"]

**Final Justification:**

Thanks to the author for the reply. Most of my concerns have been resolved. I would like to raise the score.

**Limitations:**

The rationale behind different deformable convolution designs in encoder and decoder is unclear.

The results mainly show improvements on static backgrounds rather than dynamic objects, which contradicts the main goal.

There is no ablation study to isolate the contribution of the event stream.

**Quality:**

3

**Strengths And Weaknesses:**

Strengths:
1. This work is likely the first to utilize event images specifically for depth completion with dynamic environments.
2. The proposed method is both reasonable and effective.
3. The paper is well-written and easy to follow.
4. Additionally, relevant datasets are provided.

Weaknesses:
1. Figure 2’s layout is confusing, affecting clarity of the network structure.
2. Offset notations in equations (1), (3a), and (3b) are inconsistent.
3. Lack of explanation for different deformable convolution weight designs in encoder and decoder.
4. Improvements seem more on static backgrounds than on dynamic objects, which contradicts the main goal.
5. Missing evaluation of model performance without the event stream input.

---

> ### Author Rebuttal · Authors · 2025-07-28
>
> **To Weakness1 and Question1:**
>
> The rationale behind the current layout is as follows:
>
> *1. Depth branch (S) as the main stream:* In depth completion tasks, the sparse depth input is typically treated as the primary modality, with RGB (I) and Event (E) streams serving as auxiliary sources. Therefore, we place the S branch at the center of the diagram to emphasize its dominant role in predicting the dense depth map.
>
> *2. Symmetry with the decoder:* The depth features extracted from the S branch are progressively refined through the decoding path. To clearly illustrate this symmetric encoder-decoder structure, we align the S branch horizontally with the decoder layers.
>
> *3. Cross-modal guidance via EMA modules:* The EMA modules fuse features from the I and E branches to enhance the S branch at multiple stages. This hierarchical guidance flows into the next layer of the S branch. Maintaining the current top-down layout (I at top, S in center, E at bottom) allows us to clearly visualize this progressive refinement process.
>
> According the reviewer's suggestion, we do explore alternative layouts, including swapping the positions of the S and E branches. However, we find that these configurations introduce more visual clutter, particularly due to increased crossing of connection lines, which makes the figure harder to follow. After comparison, we determine that the current configuration offers better visual balance and a more intuitive interpretation of the network architecture.
>
> Nevertheless, we agree that Figure 2 can benefit from improved clarity and thank the reviewer for this suggestion. We will revise the figure with a cleaner layout and clearer labeling to enhance readability and ensure that the design intentions are more easily understood.
>
> **To Weakness2 and Question2:**
>
> We will adopt Δ𝑝 as the unified notation for the offset throughout the paper. This would help avoid confusion and improve the overall readability of the mathematical formulations.
>
> **To Weakness3, Question3, and Limitation1:**
>
> As shown in Equation (1), the weight term of the deformable convolution in DCNv2 is represented as 𝑤·Δ𝑚, where 𝑤 denotes the learnable convolution kernel and Δ𝑚 is the modulation scalar that adjusts the importance of each sampled position in the offset domain. The *only difference* in the deformable convolution weights between the encoder and decoder lies in the treatment of Δ𝑚:
>
> In the encoder, Δ𝑚 is set as an identity matrix and does not participate in learning;
>
> In the decoder, Δ𝑚 is adaptively learned based on event information.
>
> The reasons for such design are as follows:
>
> As illustrated in Fig. 2, the EMA modules in the encoder primarily adjust the pixel distributions of RGB and depth features individually using event information. They further align the RGB-D features under the constraint of Equation (6), which helps mitigate misalignment caused by fast ego-motion. In other words, EMA focuses on learning the offset. Furthermore, 𝑤 is automatically learned by the network and Δ𝑚 is fixed as a non-learnable identity matrix.
>
> The LDF of the decoder pays more attention to depth recovery around dynamic object regions after EMA-based alignment. As a result, it not only learns the offsets near moving objects but also aims to learn the contribution of each position through Δ𝑚. This helps regulate the weighted summation process and promotes more accurate depth estimation around dynamic areas.
>
> At the code level, making Δ𝑚 learnable in the encoder is straightforward to implement. As shown in Table R2, two sets of comparison experiments are conducted based on the reviewer's suggestion.  The results indicate that a learnable Δ𝑚 in the encoder has little impact on performance. We thank the reviewer for this valuable suggestion and will clarify this point in the final paper.
>
> *Table R2: RMSE Results on EventDC-Real.*
> |Δ𝑚|1st| 2nd|3rd|average|
> |---|:---:|:---:|:---:|:---:|
> | non-learnable | 574.0 | 576.3 | 575.8 | 575.4 |
> | learnable        | 574.6 | 573.5 | 577.2 | 575.1 |
>
>
> **To Weakness4, Question4, and Limitation2:**
>
> *1. Focusing on the EventDC-Real dataset (i.e., Fig. 4):*
>
> The event stream reveals that dynamic regions include not only the truck but also background elements such as shaking trees. Depth unreliability often occurs both within these dynamic areas and at their boundaries with the background. As shown in Fig. 4, our proposed method gives better results in event-indicated areas and their transitions to the background. We will clarify that this outcome aligns well with the main objective of our work. Moreover, we will follow the reviewer’s suggestion to include more representative visualization results on EventDC-Real to further highlight improvements in dynamic regions.
>
> *2. Considering the full range of datasets (i.e., Figs. 4, 5, 6, and 7(b)):*
>
> The dynamic car in Fig. 5, the moving person and motorcycle in Fig. 6, and the person, bicycle, and car in Fig. 7(b) all demonstrate that our method consistently performs better around dynamic objects. These results strongly support that our method aligns well with the main goal of enhancing depth estimation in dynamic scenes.
>
>
> **To Weakness5, Question5, and Limitation3:**
>
> As shown in Table 6, we ablate our EventDC by comparing its performance with (EventDC-iv) and without (EventDC-ii) the event stream. The corresponding RMSEs are 638.3 mm and 685.9 mm, respectively. This indicates **a reduction of 47.6 mm when incorporating the event stream**. The result demonstrates the effectiveness of leveraging event information.
>
> Following the reviewer's suggestion, we construct EventDC-x by removing only the event stream from EventDC-ix, while retaining both the EMA and LDF modules with deformable convolutions. As reported in Table R3, excluding the event stream increases the RMSE and MAE by 15.2 mm and 7.5 mm, respectively. This further confirms the contribution of the event stream in enhancing depth estimation accuracy.
>
> *Table R3: More ablations on EventDC-Real.*
> | Method       | RMSE (mm) | MAE (mm) |
> |---|:---:|:---:|
> | EventDC-ix (w/ event)  | 574.0  | 179.0 |
> | EventDC-x (w/o event) | 599.2  | 196.5 |
>
> *We thank the reviewer for this suggestion, and will include the ablation study in our final paper.*

---

> ### Comment · Reviewer_1o7Y · 2025-08-04
>
> Thanks to the author for the reply. Most of my concerns have been resolved. I would like to raise the score.

---

> > ### Author Response · Authors · 2025-08-04
> > **Appreciation for the Reviewer’s Encouraging Feedback!**
> >
> > We sincerely thank the reviewer for the encouraging feedback. We are pleased that our responses have addressed most of the concerns, and we will revise the paper accordingly to further improve its quality! Thanks again!

---

### Official Review · Reviewer_aJhM · 2025-07-04

**Clarity:** 2
**Significance:** 3
**Originality:** 3
**Rating:** 4
**Confidence:** 3

**Summary:**

This paper presents a novel event-driven dynamic scene depth completion pipeline. The core research question is how to effectively predict dense depth from event camera-based sparse measurements in highly dynamic environments. This task is particularly difficult because object movement can create severe artifacts on RGB cameras and LiDAR measurements. To address these issues, this work introduces EventDC, which leverages the high temporal resolution and motion sensitivity of event cameras.  Its key design incorporates two main components: Event-Modulated Alignment (EMA) and Local Depth Filtering (LDF). Crucially, the paper also establishes the first event-based depth completion benchmark, comprising real-world, semi-synthetic, and fully synthetic datasets. The evaluation shows that EventDC achieves competitive efficiency and accuracy.

**Questions:**

- Is it possible to adjust the network architecture to trade-off between accuracy and latency to fit different deployment environments?

**Ethical Concerns:**

["NO or VERY MINOR ethics concerns only"]

**Final Justification:**

The author has sufficiently addressed my primary concerns around the pipeline latency. The reported latency of latency of 41.5 ms seems reasonable for the targeted application. The author has also recognized several key limitations of the current pipeline. My final rating score remain unchanged.

**Limitations:**

The paper acknowledges that while EventDC achieves promising results, its Event-Modulated Alignment and Local Depth Filtering modules introduce additional computational costs. This could potentially limit deployment on resource-constrained devices. While EventDC maintains competitive efficiency, its inference time is 41.5 ms. This computational latency directly determines the maximum frequency at which the system can generate depth inferences in a real-time setting. However, the paper does not explicitly discuss the implications of network latency on the overall system's real-time performance, nor does it detail how such latency might affect the amount of event information available for processing during different inference cycles.

**Paper Formatting Concerns:**

- In Table 1, the Color Camera of the first row is FILR BFS-U3-31S4C, this is likely a misspelling of FLIR BFS-U3-31S4C.
- Line 236, Promoting -> Prompting

**Quality:**

3

**Strengths And Weaknesses:**

Strengths:
- The proposed pipeline addresses depth completion in the dynamic environment. This is very important for real-world applications such as autonomous vehicles and augmented reality.
- The source code and datasets are promised to be released.

Weaknesses:
- My main concern is on the latency of the pipeline. The proposed work targets a highly dynamic environment, but the proposed pipeline takes >40ms to run based on Table 5.

---

> ### Author Rebuttal · Authors · 2025-07-28
>
> **1. Weakness (pipeline latency) and Question (trade-off design):**
>
> We fully agree that the task of predicting dense depth from event camera-based sparse measurements in highly dynamic environments is *particularly difficult*. Our paper proposes a framework to solve this very challenging task for the first time. We hope our work serves as a foundation for future research and encourages continued advancements across multiple aspects.
>
> Currently, our method primarily focuses on addressing **RGB-D misalignment** and improving **depth accuracy around moving objects**. However, we acknowledge that the present latency (41.5 ms) remains insufficient for the demands of highly dynamic scenarios. Following the suggestion of the reviewer, we explored several trade-off designs by reducing the channels of EventDC to half or one-quarter at each encoder stage, and by replacing the encoder with the lightweight EfficientNet-B0. The results in Table R1 show significantly reduced latency at the trade-off of some increase in error. Nonetheless, the performance remains superior to the suboptimal Prompting method.
>
> We thank the reviewer for this valuable suggestion, and we will incorporate these additional results in our final paper.
>
> *Table R1: Complexity comparisons on EventDC-Real.*
> | Method          | Param. (M) | Time (ms) | RSME (mm) |
> |---|:---:|:---:|:---:|
> | DySPN            | 26.3 | 9.8     | 700.1 |
> |Prompting        |326.9 | 39.5  | 670.7 |
> | ***Raw EventDC***   | 43.2  | 41.5  | **574.0** |
> | *½ channel*        | 22.8  | 25.4  | 613.3  |
> | *¼ channel*        | **11.7**  | 11.2  | 631.6  |
> | *EfficientNet-B0* |12.8   | **8.3**   |  617.5 |
>
> **2. Limitation:**
>
> Yes, the current latency of 41.5 ms (\~20.1 FPS) on EventDC-Real does not meet the real-time threshold. Since the color camera in EventDC-Real operates at 30 FPS, our model currently lags in a temporal gap of around 10 frames or approximately 300 ms of unprocessed event information. Moreover, a frame rate closer to 60 FPS (\~16.6 ms latency) is often desirable for more responsive performance in high-dynamic environments. We will include Table R1 in the final paper to improve clarity. In future work, we plan to explore more efficient architectural designs. We will also investigate optimization strategies to reduce latency and move closer to real-time performance.
>
> **3. Paper Formatting Concerns:**
>
> We thank the reviewer for checking our paper meticulously. We will correct the typos (Table 1 *FLIR BFS-U3-31S4C*, Line 236 *Prompting*), proofread and spellcheck the paper in the final version.
>
> *Finally, please let us know if you have any further questions or suggestions that could help us improve the paper!*

---

> > ### Comment · Reviewer_aJhM · 2025-08-05
> >
> > Thank you for the response. My concerns have been addressed.

---

> > > ### Author Response · Authors · 2025-08-05
> > >
> > > We are pleased that we have addressed your concerns. Thanks for your valuable and prompt feedback. Good night～

---

### Decision · Program_Chairs · 2025-09-17

**Decision:**

Accept (poster)

**Comment:**

This paper presents a novel event-driven dynamic scene depth completion pipeline which effectively predicts dense depth from event camera-based sparse measurements in highly dynamic environments.

Strengths:
1. The first work for event-driven dynamic scene depth completion.
2. Good EventDC benchmark and three useful datasets.

Weaknesses:
1. Some writing issues, such as image layout and inconsistent formula symbols.
2. Lack of explanation of dynamic backgrounds process and different deformable convolution weight designs in encoder and decoder.
3. Lack of latency analysis data

After rebuttal, the final rates are 3 Borderline accepts and 1 Strong Accept. The reviewer's doubts and questions were well addressed in the rebuttal process, and received positive comments and recognition. Given these positive reviews, the AC recommends acceptance and suggests the authors to fix some formatting issues based on the reviewers' comments.